# FALCON: Fine-grained Activation Manipulation by Contrastive Orthogonal Unalignment for Large Language Model

**Jinwei Hu, Zhenglin Huang, Xiangyu Yin, Wenjie Ruan,**

**Guangliang Cheng, Yi Dong[†], Xiaowei Huang[†]**

School of Computer Science and Informatics, University of Liverpool, UK

## Abstract

Large language models have been widely applied, but can inadvertently encode sensitive or harmful information, raising significant safety concerns. Machine unlearning has emerged to alleviate this concern; however, existing training-time unlearning approaches, relying on coarse-grained loss combinations, have limitations in precisely separating knowledge and balancing removal effectiveness with model utility. In contrast, we propose **F**ine-grained **A**ctivation manipu**L**ation by **C**ontrastive **O**rthogonal u**N**alignment (FALCON), a novel representation-guided unlearning approach that leverages information-theoretic guidance for efficient parameter selection, employs contrastive mechanisms to enhance representation separation, and projects conflict gradients onto orthogonal subspaces to resolve conflicts between forgetting and retention objectives. Extensive experiments demonstrate that FALCON achieves superior unlearning effectiveness while maintaining model utility, exhibiting robust resistance against knowledge recovery attempts. Our implementation is available at: `https://github.com/CharlesJW222/FALCON/tree/main`.

## 1 Introduction

Recent advancements in generative AI [1, 17], powered by Parameter-Efficient Fine-Tuning (PEFT) techniques, have enabled LLMs to internalize linguistic knowledge and excel across diverse tasks [3, 29]. While these models gain their capabilities from massive datasets, this reliance on large-scale corpora creates significant risks: harmful, biased, or sensitive information can become encoded and amplified, resulting in ethical violations, regulatory noncompliance, and potential misuse [28, 77, 43].

Existing mitigation strategies, such as LLM guardrails [13] or training models with expertly curated datasets to refuse harmful queries [60], are computationally expensive and often inadequate against adversarial attacks [85]. In contrast, while retraining an entire model on a cleaned dataset to eliminate harmful impacts is theoretically feasible, it is prohibitively resource-intensive for modern LLMs [44]. Additionally, adversaries can exploit PEFT to reintroduce such unwanted information, highlighting the urgent need for more effective and scalable solutions for publicly accessed LLMs [63].

To solve harmful or sensitive information in machine learning models, Machine Unlearning (MU) has emerged as a promising solution, supported by growing regulations such as the "right to be forgotten" under the GDPR [67]. It commonly developed in the non-LLMs domain and has proven effective at

---

[†] Corresponding authors: {yi.dong, xiaowei.huang}@liverpool.ac.uk

39th Conference on Neural Information Processing Systems (NeurIPS 2025).

removing specific data influences while preserving model performance [53, 8, 82]. When transferred to maintain responsible LLMs, MU offers significant advantages, being far more computationally efficient than full retraining. Unlearned models also exhibit greater inherent safety, as they lack the undesired knowledge necessary for malicious behaviors [27, 50].

Despite its potential, LLM unlearning still faces several fundamental **issues**: (**I1**) existing approaches typically rely on empirical methods like grid search to identify intervention parameters, lacking efficient and interpretable guidance within deeper LLM architectures, (**I2**) current methods normally rely on *coarse-grained* manipulation (using simplistic loss combinations that induce random representation dispersion with uncontrolled gradient dynamics, struggling to balance knowledge removal and utility preservation) rather than *fine-grained* representation manipulation (achieving more effective knowledge separation through targeted representation modification and regulated gradient dynamics for reducing damage to model utility), and (**I3**) knowledge recovery methods such as jailbreaking attack can recover the undesired information from the unlearned model [70].

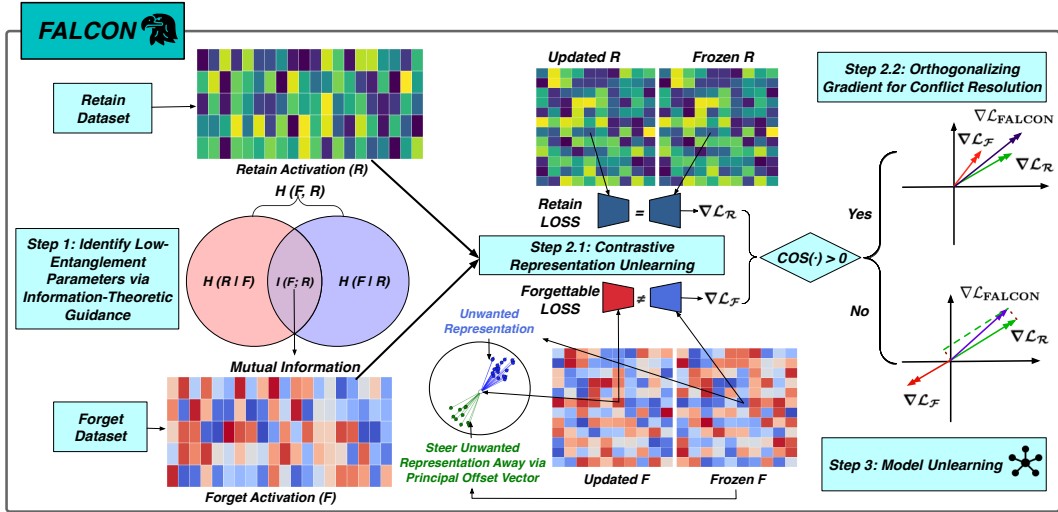

Figure 1: Schematic overview of FALCON. The pipeline comprises three stages: parameter selection based on mutual information (Step 1); contrastive orthogonal unalignment, which consists of contrastive mechanism on both forgetting and retention datasets (Step 2.1) and orthogonal gradient conflict resolution (Step 2.2); and model unlearning guided by these components (Step 3).

To address the aforementioned issues of selective knowledge unlearning in LLMs, we propose **Fine-grained Activation manipuLation by Contrastive Orthogonal uNalignment (FALCON)**, a representation-guided framework for targeted knowledge removal with minimal impact on general capabilities. For **I1**, FALCON uses mutual information (MI) as an auxiliary signal to assess dependencies between forget and retain data, based on which it introduces two core mechanisms for fine-grained disentanglement and unlearning (Step 1). To tackle **I2**, FALCON utilizes singular value decomposition (SVD) to identify principal directions in activation space to steer representations along axes misaligned with forgettable knowledge, enabling more thorough removal (Step 2.1). Meanwhile, FALCON uses a gradient orthogonal projection strategy, which constrains updates away from retention-sensitive directions, reducing interference with preserved content (Step 2.2). These mechanisms enable precise unlearning with limited data access and remain effective even under single-layer interventions. Afterwards, the projected gradients are used to update the model parameters (Step 3). For **I3**, we provide comprehensive empirical evidence and analysis in Section 5.3 and Appendix E.6 to support our claims. Our contributions are as follows:

- We propose **FALCON**, a representation-guided framework that combines contrastive mechanisms and gradient projection to achieve *fine-grained representation unalignment* in LLMs.
- We introduce **information-theoretic metrics** for quantifying knowledge entanglement, enabling principled parameter selection and providing empirical insights into knowledge distribution across model architectures.
- We demonstrate the **scalability**, **effectiveness**, and **resistance to knowledge recovery** of FALCON through extensive experiments, highlighting its ability to unlearn selective knowledge while preserving utility across various LLMs.

## 2 Related work

Our paper focuses on LLM unlearning for undesired knowledge, information-theoretic metrics, and contrastive learning. We highlight the developments and limitations of LLM unlearning in this section, while related advancements in information-theoretic metrics, contrastive learning, and gradient projection are detailed in the Appendix A and B.

**LLM Unlearning** LLM unlearning refers to the selective removal of specific knowledge from large language models while preserving their overall functionality [87]. Current approaches can be broadly categorized into training-time methods and inference-time methods [5]. Among training-time approaches, which represent the mainstream methodology, two primary directions have emerged. The first direction focuses on gradient optimization [84, 38, 18, 91, 20], which suppresses harmful knowledge through loss-driven techniques but often causes catastrophic forgetting and instability when distributions are highly similar or lack fine-grained knowledge localization. The second direction emphasizes representation-guided adaptation, targeting intermediate hidden representations for modification [50, 95, 68], but relying on empirical layer selection and lacking targeted separation mechanisms. While these aforementioned training-time methods achieve permanent unlearning by targeting specific layers and parameters, they currently rely heavily on coarse-grained loss combinations that struggle to disentangle deeply embedded knowledge representations flexibly [40].

Inference-time methods offer alternative approaches like task vectors and model editing. Task vector approaches address efficiency concerns through arithmetic operations on parameter-efficient modules, enabling lightweight unlearning under resource constraints [36, 88], but oversimplify knowledge structure through linear assumptions that fail to capture complex knowledge entanglement. In contrast, model editing usually modifies intermediate hidden states or logits to alter model behavior [5, 39, 15, 35], such as contrastive decoding methods that prevent inappropriate responses [94]. Moreover, ECO [51] has also demonstrated promising performance, though it functions more as a guardrail's definition for filtering sensitive content [14, 33], rather than directly serving as an unlearning algorithm [1] [55]. However, these methods' dependence on modular arithmetic operations fundamentally limits their granularity in knowledge separation and constrains generalizability across diverse scenarios. Additionally, in-context unlearning has emerged as another inference-time approach, leveraging tailored prompts to dynamically suppress undesired outputs [93, 62]. While flexible, this method's effect remains inherently temporary as the undesired knowledge persists in the model's representation space [54].

Despite these advancements, existing training-time methods fall short in achieving precise knowledge disentanglement between information to be forgotten and retained. To address these limitations, we propose FALCON, a targeted representation unalignment approach that achieves more precise separation through contrastive learning, gradient projection, and information-theoretic guidance. Through its contrastive mechanism and gradient projection, our approach enables fine-grained knowledge separation and resolves optimization conflicts between forgetting and retention objectives, while enhanced resistance compared to current state-of-the-art training-time methods.

## 3 Problem Formulation

### 3.1 Problem Setup

The task of LLM unlearning involves selectively removing specific knowledge (*forget set*) from the model while retaining critical information (*retain set*). However, this process is complicated by the issue of *knowledge entanglement*, where representations of the forget and retain sets overlap significantly within the model's parameters [89]. This entanglement arises due to the distributed nature of knowledge across multiple layers and features, making it difficult to isolate knowledge for removal without affecting retained information. To formalize the unlearning process, we adopt the general formulation proposed by Liu et al. [54]:

$$\min_{\theta} \left\{ \mathbb{E}_{(x,y_f) \in \mathcal{D}_{\mathcal{F}}} \left[ \mathcal{L}(y_f|x;\theta) \right] + \lambda \mathbb{E}_{(x,y) \in \mathcal{D}_{\mathcal{R}}} \left[ \mathcal{L}(y|x;\theta) \right] \right\} \tag{1}$$

where $\mathcal{L}(y|x;\theta)$ measures the discrepancy between the model's prediction and the target response $y$ for a given input $x$ under the model's parameters $\theta$. Here, $\mathcal{D}_{\mathcal{F}}$ and $\mathcal{D}_{\mathcal{R}}$ denote the forget set and retain

---

[1]Further discussion on ECO is shown in Appendix. F.3

set, respectively. The variable $y_f$ specifies the intended output for the forget set after unlearning, while the hyperparameter $\lambda \geq 0$ controls the trade-off between forgetting and retention objectives. For simplicity, we will refer to this objective as $\min_\theta \mathbb{E}_{\text{MU}}(\theta)$ in subsequent sections.

Despite the generality of above formulation, it does not explicitly quantify the representations of forgotten and retained knowledge. This lack of quantification poses challenges in precisely guiding the unlearning process [66]. To address this, a principled metric is needed to evaluate and minimize knowledge entanglement, ensuring that unlearning primarily affects the forget set while minimizing interference with the retain set. Consequently, we introduce *information-theoretic measures*, specifically continuous entropy and mutual information, to quantify the dependency between the activations of the forget and retain sets. Let $\mathcal{F}$ and $\mathcal{R}$ represent the activations of the forget and retain sets at a specific layer of the model, respectively. The degree of knowledge entanglement between representations can be formulated as the MI $I(\mathcal{F}; \mathcal{R})$:

$$I(\mathcal{F}; \mathcal{R}) = H(\mathcal{F}) + H(\mathcal{R}) - H(\mathcal{F}, \mathcal{R}) \tag{2}$$

where $H(\mathcal{F})$ and $H(\mathcal{R})$ are the continuous entropies of the activations $\mathcal{F}$ and $\mathcal{R}$, and $H(\mathcal{F}, \mathcal{R})$ denotes their joint entropy. These measures provide a systematic approach to identify parameters with minimal entanglement and guide the LLM unlearning process. The details of these metrics are shown in Appendix C.

### 3.2 LLM unlearning with MI Guidance

To quantify knowledge entanglement during machine unlearning, we use MI to measure the dependency between the activations of the forget set $\mathcal{F}^{(l)}$ and the retain set $\mathcal{R}^{(l)}$ at each layer $l$. The MI $I(\mathcal{F}^{(l)}; \mathcal{R}^{(l)})$ serves as an indicator to guide the unlearning process by minimizing entanglement between $\mathcal{F}^{(l)}$ and $\mathcal{R}^{(l)}$. To minimize the entanglement between the forget and retain sets' representations, we formulate the parameter selection for specific LLM layers as:

$$l^* = \arg\min_l I(\mathcal{F}^{(l)}; \mathcal{R}^{(l)}) \tag{3}$$

Given the selected layer $l^*$, the LLM unlearning problem guided by MI can be reformulated as:

$$\min_\theta \mathbb{E}_{\text{MU}}(\theta) \quad \text{subject to} \quad \text{Eqs. (3)} \tag{4}$$

This formulation ensures that the unlearning process is conducted on the parameters with minimal knowledge entanglement, effectively suppressing the undesired knowledge while reducing interference with the retained knowledge.

## 4 Methodology

To address the challenges of more thorough selective multi-domain knowledge unlearning and enhanced robustness against knowledge recovery in LLMs, we propose FALCON shown in Figure 1 and Appendix. D.1, a framework that advances both precision and effectiveness in knowledge manipulation. Unlike prior approaches that rely on coarse-grained loss combinations, FALCON introduces three key mechanisms: (1) mutual information-based guidance to identify parameters where knowledge representations are least entangled, enabling interpretable parameter selection; (2) contrastive mechanism with enhanced representation separation to achieve fine-grained knowledge manipulation while ensuring robust resistance against knowledge recovery attempts; and (3) gradient orthogonal projection to resolve optimization conflicts and ensure training stability. This holistic design enables precise, interpretable, and robust knowledge unlearning in LLMs, transcending traditional loss-combination methods.

### 4.1 Information-Theoretic Guidance for Unlearning

In this paper, we utilize a principled approach to selective multi-domain knowledge unlearning in LLMs through mutual information. MI provides a natural measure of representational entanglement between the forget and retain datasets across model layers. By identifying parameters that minimize MI, we can target unlearning interventions where forget and retain representations exhibit minimal overlap, thus preserving desired knowledge while selectively removing unwanted information.

We extend this measure to the multi-domain scenario where the forget set $\mathcal{F}$ consists of multiple sub-domains $\mathcal{F}_1, \mathcal{F}_2, \ldots, \mathcal{F}_m$. Our approach quantifies two critical relationships: (1) the interaction

between each sub-domain and the retain set $\mathcal{R}$, measured by $I(\mathcal{F}_i^{(l)}; \mathcal{R}^{(l)})$ at layer $l$, where lower values indicate reduced entanglement and thus more selective unlearning; and (2) the inter-domain dependencies captured by $I(\mathcal{F}_i^{(l)}; \mathcal{F}_j^{(l)})$ for sub-domains $\mathcal{F}_i$ and $\mathcal{F}_j$ $(i \neq j)$, which characterizes potential conflicts or redundancies that may impact unlearning effectiveness.

To quantify the overall representational conflicts between the forget and retain datasets, $I(\mathcal{F}^{(l)}; \mathcal{R}^{(l)})$, and the interdependence among forgettable sub-domains, $I(\mathcal{F}_i^{(l)}; \mathcal{F}_j^{(l)})$ at layer $l$, we define the aggregate MI as $I^{(l)}$:

$$I^{(l)} = \sum_{i=1}^{m} I(\mathcal{F}i^{(l)}; \mathcal{R}^{(l)}) + \eta \sum_{i=1}^{m} \sum_{j=i+1}^{m} I(\mathcal{F}_i^{(l)}; \mathcal{F}_j^{(l)}) \tag{5}$$

where $m$ denotes the number of sub-domains in the forget set $\mathcal{F}$, and $\eta$ is a balancing coefficient that controls the relative importance of inter-domain dependencies. For each layer $l$, since the activations are high-dimensional and continuous, direct entropy calculation is infeasible [75]. Instead, we utilize Kernel Density Estimation (KDE) to approximate the underlying global data distribution, estimating continuous entropy in activation space as defined in Appendix C [79]. Specifically, we use a multivariate Gaussian kernel, which offers a smooth and flexible density estimation well-suited to high-dimensional data. The estimated probability density function for activations $\mathcal{A}$ is given by:

$$p(a) = \frac{1}{Nh} \sum_{n=1}^{N} K\left(\frac{a - a_n}{h}\right) \tag{6}$$

where $a \in \mathbb{R}^d$ represents a single sample from the activations $\mathcal{A}$, including $\mathcal{F}$ and $\mathcal{R}$, with $d$ denoting the feature dimensionality of the activations, $N$ as the number of samples, $K(\cdot)$ represents the kernel function and $h$ as the adaptive bandwidth calculated using Scott's rule [69], defined as $h = \sigma N^{-\frac{1}{d+4}}$, which is particularly suitable for high-dimensional data due to its dimensionality-based adjustment. Here, $\sigma$ is the standard deviation of the data. This adaptive bandwidth selection effectively balances bias and variance, ensuring robust density estimation for diverse activation distributions [6]. To mitigate the curse of dimensionality, we apply Principal Component Analysis (PCA), which has been widely adopted across various domains in prior work [47, 65, 71] to reduce activation dimensions before performing KDE [2], retaining at least 95% of variance to ensure minimal information loss while significantly lowering computational complexity.

Using the KDE-based entropy estimations, we approximate the overall mutual information $\tilde{I}$ at each layer based on Eq. (5). The optimal layer $l^*$ for unlearning is then determined by minimizing $\tilde{I}$:

$$l^* = \arg\min_{l} \tilde{I}^{(l)} \tag{7}$$

By identifying the layer with the lowest MI, we locate the model region where the *forget* and *retain* datasets are least entangled, minimizing the overlap between the two types of knowledge. Concurrently, this layer exhibits higher entanglement among sub-domains within the *forget* set, enabling efficient updates to shared representations across forgettable sub-domains. This dual property makes the layer an optimal target for unlearning, where parameters with minimal mutual interference are prioritized to remove undesired knowledge while more easily preserving essential and generalizable knowledge for downstream tasks.

## 4.2 Contrastive Orthogonal Unalignment

To achieve selective knowledge unlearning in LLMs, we first apply MI-guided parameter selection [2] to identify layers with minimal knowledge entanglement, which remains fixed throughout unlearning. We then devise *Contrastive Orthogonal Unalignment* through contrastive mechanisms and gradient projection, employing *alternating strategy* between forget and retain datasets to iteratively refine representations while balancing knowledge removal and retention objectives.

### 4.2.1 Contrastive Representation Unlearning

The core task of LLM unlearning is to selectively separate knowledge representations to be forgotten from those to be retained. Contrastive learning provides an effective mechanism for this task by

---

[2]Discussion on MI-guided parameter selection is shown in Appendix. F.2

learning discriminative representations through comparing similar and dissimilar samples. In our context, we leverage contrastive learning to maximize the distance between representations that should be forgotten while maintaining the coherence of retained knowledge.

To facilitate thorough unlearning, we construct Principal Offset Vectors (POVs) that steer model activations away from undesired knowledge by redirecting updated forgettable representations into subspaces intentionally misaligned with the principal directions of frozen counterparts, as identified via SVD, thereby achieving representational decoupling within the model.

Mathematically, given an activation matrix $\mathcal{H} \in \mathbb{R}^{(B \cdot L) \times D}$, where $B$ is the batch size, $L$ the sequence length, and $D$ the hidden dimension, we perform SVD to obtain the dominant principal directions $v_1, \ldots, v_K$ corresponding to the top-$K$ singular values. The POVs $\mathcal{H}^+$ is defined as:

$$\mathcal{H}^+ = \frac{f\left(r \cdot \left(I - w \sum_{i=1}^{K} v_i v_i^\top\right), \epsilon\right)}{\left| f\left(r \cdot \left(I - w \sum_{i=1}^{K} v_i v_i^\top\right), \epsilon\right) \right|} \tag{8}$$

Here, $r \in \mathbb{R}^D$ is a randomly initialized vector, $w$ controls the influence of principal directions, and $I \in \mathbb{R}^{D \times D}$ is the identity matrix. The term $\epsilon$ introduces optional perturbations while $f(\cdot)$ is a flexible transformation operator, potentially including non-linear mappings (e.g., tanh), adaptive projections, or adversarially-inspired perturbations, enhancing disentanglement and recovery resistance. This design ensures $\mathcal{H}^+$ is directed away from dominant principal subspaces, combining deterministic guidance and transformations to improve robustness. Unlike generic random vectors, POVs deliberately target dominant features to improve adversarial robustness and unlearning efficacy.

For each input sample, we define three types of representations: the anchor representation $\mathcal{H}_a$ from the updated model for the forget set, the positive representation $\mathcal{H}^+$, given by the POV defined in Eq. (8), and the negative representations $\mathcal{H}^-$ from the frozen model. To ensure consistent scaling, all representations are normalized, and their similarity scores are measured using cosine similarity:

$$S^+ = \sum_{d=1}^{D} \mathcal{H}_a[d] \cdot \mathcal{H}^+[d], \quad S^- = \sum_{z=1}^{\mathcal{Z}} \sum_{d=1}^{D} \mathcal{H}_a[d] \cdot \mathcal{H}_z^-[d] \tag{9}$$

where $\mathcal{Z}$ is the number of negative samples. Building on these similarity scores, we define the forget loss $\mathcal{L}_\mathcal{F}$ using the InfoNCE objective:

$$\mathcal{L}_\mathcal{F} = -\frac{1}{|B|} \sum_{b=1}^{|B|} \log \frac{\exp(S_b^+/\tau)}{\exp(S_b^+/\tau) + \sum_{b=1}^{\mathcal{N}} \exp(S_b^-/\tau)} \tag{10}$$

where $\tau$ is a temperature scaling parameter. This loss encourages the updated model's representations to align with the POVs while diverging from the frozen model's representations of undesired knowledge. By leveraging both directional guidance through POVs and contrastive learning, our approach achieves more precise and efficient representation unalignment in activation space.

In addition to unlearning undesired representations, preserving critical knowledge for downstream tasks is essential. We define a retain loss in Eq. (11) to measure alignment between the updated model's activations ($\mathcal{H}^u$) and frozen model's activations ($\mathcal{H}^f$) for the retain set. This retention alignment loss, functioning as a self-supervised variant of contrastive loss, maximizes consistency between updated and frozen activations to ensure effective knowledge preservation during unlearning.

$$\mathcal{L}_\mathcal{R} = 1 - \frac{1}{|B|} \sum_{b=1}^{|B|} \frac{\sum_{d=1}^{D} \mathcal{H}_b^u[d] \cdot \mathcal{H}_b^f[d]}{\sqrt{\sum_{d=1}^{D} (\mathcal{H}_b^u[d])^2} \cdot \sqrt{\sum_{d=1}^{D} \left(\mathcal{H}_b^f[d]\right)^2}} \tag{11}$$

This loss ensures alignment between the updated and frozen model activations for the retain set, preserving critical knowledge while complementing the unlearning objective. Combined with the forget loss $\mathcal{L}_\mathcal{F}$, this approach achieves an effective balance between unlearning and retention.

### 4.2.2 Orthogonalizing Gradient for Conflict Resolution

After computing the forget loss $\mathcal{L}_\mathcal{F}$ and retain loss $\mathcal{L}_\mathcal{R}$, we address optimization direction misalignment between unlearning and retaining by employing a gradient projection mechanism that orthogonalizes conflicting gradients onto subspaces, minimizing interference and promoting balanced optimization. Given the gradients of the forget and retain losses, denoted as $\nabla \mathcal{L}_\mathcal{F}$ and $\nabla \mathcal{L}_\mathcal{R}$, respectively, the conflict can be quantified using the cosine similarity:

$$cos(\nabla\mathcal{L}_{\mathcal{F}}, \nabla\mathcal{L}_{\mathcal{R}}) = \frac{\nabla\mathcal{L}_{\mathcal{F}} \cdot \nabla\mathcal{L}_{\mathcal{R}}}{\|\nabla\mathcal{L}_{\mathcal{F}}\| \cdot \|\nabla\mathcal{L}_{\mathcal{R}}\|} \tag{12}$$

where $cos(\cdot) < 0$ indicates opposing directions, signifying a conflict between the two objectives. To mitigate this conflict, we adjust the gradients by projecting one onto the orthogonal complement of the other. Specifically, if $cos(\cdot) < 0$, we project $\nabla\mathcal{L}_{\mathcal{F}}$ onto the subspace orthogonal to $\nabla\mathcal{L}_{\mathcal{R}}$:

$$\nabla\mathcal{L}_{\mathcal{F}}^{\text{proj}} = \nabla\mathcal{L}_{\mathcal{F}} - \frac{\nabla\mathcal{L}_{\mathcal{F}} \cdot \nabla\mathcal{L}_{\mathcal{R}}}{\|\nabla\mathcal{L}_{\mathcal{R}}\|^2}\nabla\mathcal{L}_{\mathcal{R}} \tag{13}$$

This adjustment ensures that $\nabla\mathcal{L}_{\mathcal{F}}^{\text{proj}}$ is orthogonal to $\nabla\mathcal{L}_{\mathcal{R}}$, eliminating interference from the retain objective during the update for the forget objective. Once the gradients are adjusted, the final update direction of the FALCON is determined by combined gradients:

$$\nabla\mathcal{L}_{FALCON} = \alpha\nabla\mathcal{L}_{\mathcal{F}}^{\text{proj}} + \beta\nabla\mathcal{L}_{\mathcal{R}} \tag{14}$$

where $\alpha$ and $\beta$ are hyperparameters balancing the contributions of the forget and retain objectives.

This mechanism mitigates gradient conflicts, enabling joint optimization while minimizing interference. By enforcing orthogonality between adjusted gradients, it approximates a Pareto-optimal solution. The model then updates its weights using the conflict-reduced gradient, allowing for more flexible adaptation. To further enhance efficiency and stability, we leverage the second-order optimizer Sophia [52], as suggested in [25, 41], for refined weight updates, ensuring a more effective and stable optimization process for selective knowledge unlearning.

## 5 Experiments

To validate FALCON's effectiveness, we conduct extensive experiments to answer the following research questions: **RQ1**: Does FALCON with MI guidance, establish a quantifiable measure for principled parameter selection while achieving superior performance in *harmful knowledge unlearning* tasks? (Section 5.1) **RQ2**: Does FALCON maintain strong generalizability across diverse unlearning tasks including *entity unlearning* and *copyrighted content unlearning*? (Section 5.2) **RQ3**: Beyond efficient parameter space reduction through MI guidance, does FALCON's algorithmic design offer competitive *computational efficiency*? (Appendix E.3) **RQ4**: Can FALCON effectively resist *recovery attempts* of unlearned knowledge? (Section 5.3). More complete experiments and ablation study are shown in Appendix E.

### 5.1 Harmful Knowledge Unlearning

To validate **RQ1**, we use the **WMDP** [50] benchmark for harmful knowledge unlearning assessment, **WikiText** [59] for measuring perplexity, and **MMLU** [26] for evaluating model utility. We test FALCON on three pre-trained LLMs: **Zephyr-7B-Beta** [76], **Yi-6B-Chat** [86], and **Mistral-7B-Instruct-v0.3** [42], comparing against all baselines from [50], with details in Appendix D.

#### 5.1.1 Mutual Information for Parameter Selection

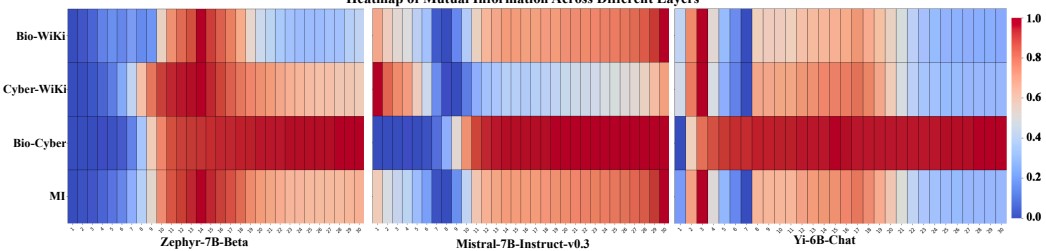

Figure 2: Heatmaps of MI across LLM layers show that lower MI values indicate layers better suited for unlearning, with early layers being more domain-specific and deeper layers more entangled.

***Visualization of MI for LLMs*** Figure 2 presents MI heatmaps illustrating knowledge entanglement between forget sets (WMDP-Bio, WMDP-Cyber) and the retain set (WikiText-2-raw-v1) across LLM layers. This metric provides an interpretable measure for identifying layers with minimal entanglement for targeted unlearning. All models show lower MI values in earlier layers, indicating more domain-specific and disentangled representations, which aligns with both intuition and experimental observations [50]. Yi-6B-Chat demonstrates particularly complex entanglement patterns between

domains, presenting a greater difficulty for unlearning multi-domain knowledge and making it an ideal candidate for our effectiveness analysis experiments in Section 5.1.2. Beyond identifying optimal intervention parameters, MI-guided selection improves efficiency by narrowing the parameter search space compared to exhaustive methods like grid search, scaling effectively with model complexity.

***Gradient Conflicts Analysis*** We empirically

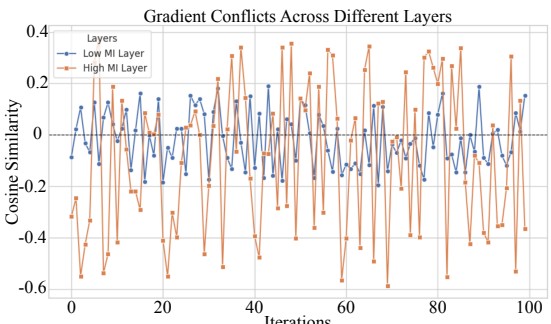

Figure 3: Gradient conflicts across layers with minimum (blue) and maximum (orange) MI values computed during parameter selection in Mistral-7B.

validate the underlying principle of MI guidance by analyzing gradient conflicts between forget and retain objectives across layers. As shown in Figure 3, layers with low MI values exhibit significantly reduced conflicts, with cosine similarities near zero, indicating minimal interference between objectives. Conversely, high-MI layers show pronounced, fluctuating conflicts, highlighting the issues of entangled representations. These results confirm that mutual information is a reliable auxiliary signal for guiding parameter selection, as low-MI parameters reduce interference, support stable updates, and help mitigate conflicts between unlearning and retention goals.

### 5.1.2 Unlearning Effectiveness and Utility Analysis

We evaluate FALCON against all baseline methods across three LLM architectures shown in Table 1 and Appendix E.1, with our evaluation focusing on three key metrics: WMDP scores for measuring unlearning effectiveness, MMLU scores for assessing general knowledge retention, and perplexity (PPL) for model stability. Our primary objective is to *minimize WMDP scores while maintaining MMLU and PPL values close to the base model's performance (MMLU and PPL)*, as this indicates successful knowledge removal without compromising general capabilities. To ensure quantifiable comparison, we prioritize maintaining general model utility and report each method's best unlearning performance under this setting. Results demonstrate FALCON's superior performance compared to baselines that struggle

Table 1: Unlearning effectiveness and utility across models and methods. Metrics with (↑) indicate preferable increases; (↓) indicate preferable decreases.

| Method | WMDP (↓) | | MMLU (↑) | PPL (↓) |
|---|---|---|---|---|
| | **Bio** | **Cyber** | | |
| Zephyr-7B | 63.7 | 43.8 | 58.1 | 1.5 |
| + LLMU | 36.3 | 40.5 | 50.3 | 4.8 |
| + SCRUB | 38.7 | 35.4 | 50.0 | 16.5 |
| + SSD | 53.1 | 43.2 | 52.8 | 1.6 |
| + RMU | 34.5 | 28.9 | 57.4 | 1.5 |
| **+ FALCON** | **26.7** | **25.3** | **57.4** | **1.5** |
| Yi-6B-Chat | 65.4 | 42.6 | 61.8 | 1.5 |
| + LLMU | 56.2 | 39.9 | 57.5 | 5.4 |
| + SCRUB | 38.7 | 35.5 | 50.0 | 16.4 |
| + SSD | 55.1 | 43.7 | 53.8 | 1.6 |
| + RMU | 50.8 | 33.5 | 59.6 | 1.6 |
| **+ FALCON** | **27.7** | **25.3** | **60.3** | **1.5** |

with effectiveness-utility balance and show increased uncertainty in their perplexity. On Zephyr-7B, FALCON achieve lower forgetting scores while preserving general capabilities. This advantage is more clear on Yi-6B-Chat with its complex knowledge entanglement: RMU show significant biological domain degradation when constrained to maintain MMLU above 60%, while FALCON maintain consistent effectiveness with superior general performance. These findings validate our fine-grained representation-guided mechanisms for targeted unlearning with preserved utility, even in scenarios with complex knowledge entanglement.

### 5.2 Cross-Domain Generalizability Assessment

To address **RQ2**, we conduct additional experiments on copyrighted content and entity unlearning using the **MUSE** [72] and **TOFU** [56] benchmarks with additional baselines [16]. For **RQ3**, we compare computational efficiency across methods in Appendix E.3. All aforementioned experiments utilize *first-order optimizers for fair comparison*, with complete implementation details in Appendix D.

### 5.2.1 Copyrighted Content Unlearning

For copyrighted content unlearning, we utilize the MUSE benchmark and Llama-2-7b-hf to assess FALCON's effectiveness in removing protected news articles while preserving general capabilities. As shown in Table 2, FALCON achieved the lowest forget metrics scores (0.02 and 0.03) while maintaining competitive retention (0.54). Unlike baselines, FALCON consistently balanced copyright removal with knowledge preservation, demonstrating broader applicability beyond harmful content removal.

Table 2: Evaluation on MUSE News over 10 epochs.

| Method | forget_knowmem_ROUGE↓ | forget_verbmem_ROUGE↓ | retain_knowmem_ROUGE↑ |
|---|---|---|---|
| Finetuned | 0.64 | 0.58 | 0.55 |
| Retain | 0.33 | 0.21 | 0.56 |
| GradAscent | 0.00 | 0.00 | 0.00 |
| GradDiff | 0.41 | 8.92e-3 | 0.37 |
| NPO | 0.56 | 0.35 | 0.51 |
| SimNPO | 0.54 | 0.36 | 0.51 |
| RMU | 0.48 | 0.05 | 0.51 |
| **FALCON** | **0.02** | **0.03** | **0.54** |

### 5.2.2 Entity Unlearning

We evaluate FALCON's ability to remove knowledge about fictitious entities using TOFU with varying forget data sizes (1/5/10%). Our method maintain strong forget quality (FQ↑) and model utility (MU↑) across different splits on Llama-3.2-1B-Instruct. Even with only 10 unlearning epochs, FALCON consistently outperform baselines in balancing knowledge removal with preserved utility. Notably, while other methods like GradAscent suffers significant utility degradation with larger forget sets, FALCON remains effective, demonstrating our method's generalizability to entity unlearning tasks.

Table 3: TOFU evaluation across varying sizes over 10 epochs.

| Method | Forget01 | | Forget05 | | Forget10 | |
|---|---|---|---|---|---|---|
| | FQ | MU | FQ | MU | FQ | MU |
| Finetuned | 0.01 | 0.60 | 2.96e-13 | 0.60 | 8.08e-22 | 0.6 |
| Retain | 1.0 | 0.60 | 1.0 | 0.60 | 1.0 | 0.59 |
| GradAscent | 0.27 | 0.33 | 1.94e-119 | 0 | 1.06e-239 | 0 |
| GradDiff | 0.77 | 0.43 | 2.04e-110 | 0.22 | 1.06e-239 | 0.49 |
| IdkDPO | 0.01 | 0.51 | 4.02e-06 | 0.04 | 4.26e-10 | 0.08 |
| NPO | 0.92 | 0.56 | 0.32 | 0.42 | 0.02 | 0.46 |
| RMU | 0.16 | 0.55 | 1.46e-7 | 0.57 | 1.4e-20 | 0.59 |
| **FALCON** | **0.99** | **0.55** | **0.92** | **0.59** | **0.52** | **0.60** |

### 5.3 Resistance Against Knowledge Recovery Attempts

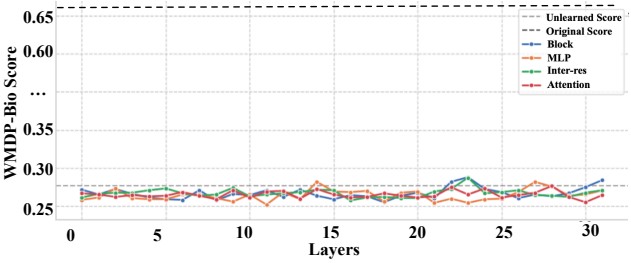

Figure 4: Logit lens probing results on different components.

We conduct experiments on Yi-6B-Chat to evaluate FALCON's resistance against knowledge recovery attempts [55] for **RQ4**. Logit Lens [61], which projects intermediate activations onto the model's vocabulary space, serves as a powerful technique for probing the model's internal knowledge representations and potential recovery of unlearned information. As shown in Figure 4, the logit lens analysis across different architectural components such as MLP and attention layers demonstrates that the unlearned knowledge remains consistently inaccessible, with performance staying close to the unlearned baseline and far below the original model's performance.

Additionally, as shown in Table 4, FALCON exhibits strong resilience against enhanced GCG in QA setting, an advanced prefix-optimization based jailbreaking attack that compromises other baselines such as RMU [73]. Even with increasing attack iterations, the recovered WMDP scores remain close to the unlearned baseline, demonstrating robust unlearning through fundamental changes to the model's internal representations rather than superficial knowledge mask-

ing. Further evaluation using conversational templates for jailbreaking attacks (detailed in Appendix E.6) further validates our method's robustness against knowledge recovery attempts.

These results across both probing techniques validate FALCON's effectiveness in creating a more permanent and recovery-resistant form of knowledge removal.

Table 4: Knowledge recovery results via enhanced GCG attack.

| Dataset | Original Score | Unlearning Score | Recovery Score via Enhanced GCG | | | |
|---|---|---|---|---|---|---|
| | | | GCG-500 | GCG-1000 | GCG-1500 | GCG-2000 |
| WMDP-Bio | 65.4 | 27.7 | 27.6 | 28.4 | 27.9 | 28.9 |
| WMDP-Cyber | 42.6 | 25.3 | 26.3 | 26.4 | 25.8 | 24.7 |

## 6 Practical Implications of LLM Unlearning for Responsible AI

The problem setting addressed by FALCON[3] stems from the growing challenge of directly employing LLMs or deploying them as autonomous agents in safety-critical environments [32, 34]. As these models become increasingly embedded in diverse real-world applications, selectively removing undesired or harmful knowledge after deployment remains difficult [54]. Unlike conventional machine learning models where unwanted data can simply be excluded in future training cycles, LLMs encode information across billions of parameters, making precise removal extremely challenging. This limitation creates a critical gap between learning capabilities and responsible deployment. The issue is further amplified by regulatory demands such as the GDPR's "right to be forgotten" [67], and by empirical evidence that even state-of-the-art LLMs and their agentic variants can inadvertently reproduce sensitive or hazardous content when prompted, raising urgent concerns about information safety and controllability.

FALCON provides a fine-grained unlearning mechanism that identifies harmful knowledge and decouples it from beneficial reasoning. This targeted process enables models to forget unsafe information while retaining legitimate competence, supporting the emerging need for responsible LLM deployment [58, 31]. As LLMs operate in dynamic, real-world contexts, the capacity for precise and interpretable knowledge modification becomes essential for responsible AI. We advocate viewing unlearning not as an academic objective but as core practical infrastructure for transparent, compliant, and responsible AI systems.

## 7 Conclusion

This paper presents FALCON, a fine-grained representation-guided framework for LLM unlearning. Leveraging mutual information guidance and contrastive orthogonal unalignment, it enables precise and efficient unlearning through principal component-based representation separation and gradient conflict resolution. Extensive experiments demonstrate its superior performance in effectively removing undesired knowledge while preserving essential information across diverse tasks, along with resistance against knowledge recovery and efficient optimization guidance. However, this work is currently limited to text-based LLM unlearning, with experiments conducted on relatively smaller models due to computational constraints. Future directions include extending unlearning to multi-modal LLMs and refining strategies to disentangle intertwined knowledge in deeper architectures.

## Acknowledgment

This work is partially funded by the European Union (under grant agreement ID 101212818). Views and opinions expressed are however those of the author(s) only and do not necessarily reflect those of the European Union or European Health and Digital Executive Agency (HADEA). Neither the European Union nor the granting authority can be held responsible for them. This work is partially supported by Innovate UK through AI-PASSPORT under Grant 10126404. Yi's contribution is partially supported through the Royal Society international exchanges programme and in part by the Engineering and Physical Sciences Research Council [grant number EP/Y009800/1], through funding from RAi UK.

---

[3]Additional discussions are provided in Appendix F.

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
