# OpenReview forum: "FALCON: Fine-grained Activation Manipulation by Contrastive Orthogonal Unalignment for Large Language Model"
_NeurIPS.cc/2025/Conference — NeurIPS 2025 poster_

### Official Review · Reviewer_amhk · 2025-06-07

**Clarity:** 3
**Significance:** 3
**Originality:** 3
**Rating:** 4
**Confidence:** 4

**Summary:**

This paper introduces Falcon, a novel machine unlearning method designed to remove specific information from large language models (LLMs) while preserving the utility of their representations.
Falcon consists of two key components:
a) Layer selection — identifying layers where the target information to retain and the information to forget are least entangled, based on mutual information analysis;
b) Representation disentanglement — applying contrastive orthogonal unalignment followed by orthogonal gradient conflict resolution to guide the model in effectively unlearning undesired information while preserving useful features.

**Questions:**

See weaknesses

**Ethical Concerns:**

["NO or VERY MINOR ethics concerns only"]

**Final Justification:**

This is a solid paper with reasonable contributions. The authors acknowledged the literature gap I pointed out, so I have maintained my fairly positive score.

**Limitations:**

yes

**Quality:**

3

**Strengths And Weaknesses:**

Strengths:

The paper is generally well-written (aside from a few sections noted below) and easy to follow.

The authors present a comprehensive experimental setup, effectively demonstrating the potential of their method.

The method is shown to be highly resilient against knowledge recovery attempts, which is an important and timely contribution.

Weaknesses and Suggestions:

The paper is not always fully self-contained. For example, the baselines are only briefly discussed in the Appendix, while the main text repeats the MI-based idea in Sections 3.2 and 4.1. It would be more helpful to consolidate the explanation of the proposed method and dedicate more space in the main paper to clearly describe the baselines and how they differ from the approach proposed here.

Several key ideas in the paper—such as removing unwanted information using SVD or PCA, or preserving desired information while discarding irrelevant components (as in Equation 9)—have been explored in prior work that is not cited. Specifically, related approaches appear in the following references:

[1] https://arxiv.org/pdf/2203.07893

[2] https://proceedings.mlr.press/v206/kleindessner23a/kleindessner23a.pdf

[3] https://proceedings.neurips.cc/paper_files/paper/2024/file/684c59d614fe6ae74a3be8c3ef07e061-Paper-Conference.pdf

I’m not suggesting that these works necessarily undermine the novelty of your method, but a more thorough discussion of the similarities and differences would help clarify your contribution and position it more clearly within the existing literature.

---

> ### Author Rebuttal · Authors · 2025-07-29
>
> We sincerely appreciate your feedback and support for our work. Regarding your remaining concerns and suggestions, we are pleased to adopt them.
>
> $\textbf{For Weakness and Suggestions 1: }$
>
> Regarding the repetition between Sections 3.2 and 4.1, we clarify our organizational rationale: Section 3.2 establishes the formal problem formulation with mutual information as the theoretical foundation for quantifying knowledge entanglement, while Section 4.1 provides concrete implementation details including KDE-based estimation and multi-domain extension. Though intended to separate theoretical formulation from practical implementation, we acknowledge this may created redundancy and will consolidate similar equations and descriptions in the camera-ready version.
>
> Regarding baseline comparisons, we provide a detailed discussion below that will be integrated into the Related Work section and Appendix in the final version.
>
> Existing LLM unlearning approaches operate through distinct methodological paradigms with inherent trade-offs. Parameter-space methods like Task Vector [1] use arithmetic operations on model weights for computational efficiency but oversimplify knowledge structure through linear assumptions that fail to capture complex knowledge entanglement. Gradient-based approaches including LLMU [2] and GradAscent [3] apply gradient ascent to forgotten knowledge but frequently induce catastrophic forgetting and optimization instability. Preference optimization methods such as NPO and its variants reformulate unlearning as preference learning, yet are prone to catastrophic collapse when distributions exhibit high similarity and lack fine-grained knowledge localization control [4,5] . Representation manipulation methods like RMU [6] modify intermediate activations using MSE loss and random unit vectors for steering, offering more targeted intervention than gradient methods but relying on empirical layer selection and lack targeted separation. Similar to RMU, FALCON belongs to the representation-based paradigm, optimizing only a subset of intermediate layers and parameters for effective LLM unlearning. However, FALCON advances through three innovations: information-theoretic parameter guidance using mutual information to identify optimal intervention layers, targeted representation disentanglement via contrastive learning with Principal Offset Vectors for precise knowledge separation, and gradient orthogonal projection for conflict resolution between objectives. Compared to RMU's MSE-based random steering with heuristic selection, our contrastive mechanism provides more principled and targeted representation manipulation, achieving superior unlearning effectiveness while preserving model utility.
>
> $\textbf{For Weakness and Suggestions 2: }$
>
> These suggested works indeed present important insights that share fundamental concepts with our approach. SEA [7] demonstrates how SVD on cross-covariance matrices can project LLM activations toward directions with maximal covariance with positive demonstrations while minimizing covariance with negative ones for inference-time behavior modification. Efficient Fair PCA [8] provides an elegant solution for fair dimensionality reduction by ensuring projected data's group-conditional means coincide while maintaining computational efficiency. SAL [9] introduces a clever approach applying SVD to cross-covariance matrices between representations and protected attributes, pruning high-covariance directions to remove sensitive information with superior computational efficiency. While all these works share the fundamental insight of using SVD and PCA to manipulate information in neural representations efficiently, each addresses distinct challenges with novel technical contributions.  We will certainly include proper citations to these works in our methodology section of the camera-ready version to acknowledge how similar techniques have been developed and applied across the literature.
>
> $\textbf{Reference List:}$
>
> [1] llharco et al., "Editing Models with Task Arithmetic". ICLR 2023.
>
> [2] Yao et al., "Large Language Model Unlearning". Neurips 2024
>
> [3] Jang et al., "Knowledge Unlearning for Mitigating Privacy Risks in Language Models". ACL 2023.
>
> [4] Zhang et al., "Negative Preference Optimization: From Catastrophic Collapse to Effective Unlearning", COLM 2024
>
> [5] Fan et al., "Simplicity Prevails: Rethinking Negative Preference Optimization for LLM Unlearning", Neurips SafeGenAi 2024
>
> [6] Li et al., "The WMDP benchmark: Measuring and reducing malicious use with unlearning". ICML 2024
>
> [7] Qiu et al., "Spectral Editing of Activations for Large Language Model Alignment". NeurIPS 2024
>
> [8] Kleindessner et al., "Efficient fair PCA for fair representation learning". AISTATS 2023
>
> [9] Shao et al., "Gold Doesn’t Always Glitter: Spectral Removal of Linear and Nonlinear Guarded Attribute Information". EACL 2023

---

> > ### Comment · Reviewer_amhk · 2025-08-04
> >
> > I thank the authors for replying.
> > Please include the mentioned works in your manuscript, those are just a few I'm familiar with, I'm sure there are others out there.

---

> ### Author Response · Authors · 2025-08-04
>
> Thank you for your timely feedback and suggestions. We will include the mentioned works in our manuscript and identify compressive relevant references to strengthen our literature review.

---

### Official Review · Reviewer_ASj8 · 2025-06-26

**Clarity:** 2
**Significance:** 3
**Originality:** 3
**Rating:** 4
**Confidence:** 2

**Summary:**

This paper introduces FALCON, a novel framework for accurate and robust knowledge de-learning in LLMs. Unlike previous works that rely on retraining, parameter overwriting, or on-the-fly tuning, FALCON is a representation-guided framework that combines contrastive mechanisms and gradient projection to achieve fine-grained representation de-learning in LLMs.
Experiments on multiple de-learning tasks on multiple open source LLMs show its good performance.

**Questions:**

See above.

**Ethical Concerns:**

["NO or VERY MINOR ethics concerns only"]

**Final Justification:**

I have responsed to author. This is my final Justification

**Limitations:**

yes

**Quality:**

3

**Strengths And Weaknesses:**

Strengths:

1.	This paper focus on knowledge unlearning, which is a high-impact problem, particularly in light of privacy and safety regulations. FALCON leverages mutual information guidance and contrastive orthogonal unalignment, it enables precise and efficient unlearning.

2.	This method explicitly tackles the common trade-off between forgetting and preserving useful knowledge, which is often ignored or vaguely addressed in prior work.
3.	The paper demonstrates FALCON’s effectiveness in various settings, including robustness to jailbreak-style recovery attacks, and generalization across tasks and model sizes.

Weaknesses:

1.	While MI-guided layer selection is compelling, the paper does not quantify its cost in practice, especially for large models or datasets. It’s unclear whether this process is scalable.
2.	Since FALCON relies on internal activation and gradient access, it is not applicable to closed-source models like GPT-4 or Claude. It is good to include discussion or experiments on how FALCON could be adapted or approximated for black-box models or low-resource settings.
3.	The method hinges on identifying principal directions of harmful representations, but it is not evaluated how stable this direction is under perturbations or data sampling variance.

---

> ### Author Rebuttal · Authors · 2025-07-29
>
> We thank you for your feedback and support of our work. We are very happy to provide additional experiments and discussion to address your remaining concerns.
>
> $\textbf{For Weakness 1:}$
> For the computational cost and scalability of MI-guided parameter selection, we additionally provide a detailed cost analysis below to demonstrate the practical feasibility of our design.
>
> MI Computational Cost Analysis (TOFU + Llama-3.2):
>
> Sample Size | Time(s)
> ------------|--------
> 10%         | 67
> 30%         | 79
> 50%         | 110
> 70%         | 165
> 100%        | 260
>
> The results show reasonable results of only 4.3 minutes for the complete TOFU dataset [1]. This represents manageable overhead as a one-time preprocessing step. Importantly, our MI-guided approach eliminates the need for costly empirical grid search across all model layers that traditional methods require, making our approach both theoretically principled and computationally practical for larger models and datasets. Additional discussion can be found in Review2, Q1. Furthermore, our Appendix E.3 provides computational cost comparison of our method, which demonstrates FALCON's competitive efficiency compared to baseline methods, indicating the scalability of our approach.
>
> $\textbf{For Weakness 2:}$
> We understand your interest in exploring black-box LLM unlearning methodology. Actually, FALCON is a white-box algorithm, which requires access to internal activations and also gradients. However, FALCON's core design principles could be possibly adapted for black-box scenarios through contrastive prompt engineering that mirrors our contrastive orthogonal unalignment mechanism. Building upon recent advances in in-context unlearning [2,3], this adaptation could leverage surrogate models, smaller accessible models trained to approximate the behavior of target closed-source systems [4], to identify principal directions of unwanted knowledge representations through our SVD-based analysis, then systematically design prompts that incorporate counter-examples and directional guidance that implement our Principal Offset Vectors (POVs) concept at the prompt level. The information-theoretic principles underlying our mutual information calculations could provide crucial guidance for optimizing such unlearning prompts by quantifying the entanglement between different knowledge domains within the prompt structure itself, enabling systematic optimization of prompt templates that maximize separation between forget and retain domains within API-only constraints.
>
> $\textbf{For Weakness 3:}$
> We understand your concern about principal direction design. To address this concern, we provide comparative experiments between random perturbation-based baselines and our method under identical knowledge recovery attempts.
>
> This observation actually highlights a key advantage of our approach. While RMU [5] relies on random perturbations which are inherently unstable and susceptible to variance, our method uses Principal Component Analysis to identify the dominant subspaces of harmful representations through SVD decomposition. This principled approach provides significantly more stable directions compared to random perturbations. Our empirical results under identical jailbreaking attempts demonstrate this stability advantage through comparative evaluation showing that our method maintains consistent performance with minimal variance, while RMU's knowledge can be substantially recovered under identical conditions. Additionally, our logit lens results (Section 5.3, Figure 4) also demonstrate that our method is more stable compared to the logit lens visualization results of other baselines in [4].
>
> Method        |Original|Unlearned|Recovery
> --------------|--------|---------|---------------
> RMU (Bio)     |65.4    |50.8     |58.5 ± 3.2
> FALCON (Bio)  |65.4    |27.7     |28.1 ± 0.5
> RMU (Cyber)   |42.6    |33.5     |41.8 ± 2.1
> FALCON (Cyber)|42.6    |25.3     |25.5 ± 0.8
>
> These results demonstrate that our SVD-based principal directions are inherently more stable than random vector approaches. This stability stems from the mathematical properties of principal components, which capture the most significant variance directions in the data rather than arbitrary perturbations.
>
> $\textbf{Reference List:}$
>
> [1] Maini et al., "TOFU: A task of fictitious unlearning for LLMs". COLM 2024
>
> [2] Pawelczyk et al., 2024 - "In-context unlearning: Language models as few-shot unlearners". ICML2024
>
> [3] Zheng et al., 2023 - "Can we edit factual knowledge by in-context learning?". EMNLP2023
>
> [4] Pan et al., 2024. "Differential privacy in deep learning: A literature survey". Neurocomputing
>
> [5] Li et al., "The WMDP benchmark: Measuring and reducing malicious use with unlearning". ICML 2024

---

> ### Comment · Reviewer_ASj8 · 2025-08-06
>
> Thank you for your response. Some of concerns have been addressed. I keep a positive score(4).

---

### Official Review · Reviewer_RCfm · 2025-07-01

**Clarity:** 2
**Significance:** 3
**Originality:** 2
**Rating:** 4
**Confidence:** 4

**Summary:**

This paper focuses on large language model (LLM) unlearning with activation manipulation. Specifically, this work aims to improve the existing unlearning methods, regarding the limitations in relying on coarse-grained loss combinations. To this end, this work proposes a Fine-grained activation manipulation by contrastive orthogonal unalignment, termed FALCON, that incorporating information-theoretic guidance for efficient parameter selection, and project conflict gradients onto orthogonal subspaces to solve the onflicts between forgetting and retention objectives. Various experiments are conducted to demonstrate the performance effectiveness.

**Questions:**

1. Could the authors further elaborate what is “fine-grained” compared with “coarse-grained” in the context of LLM unlearning considered in this work?
2. Could the authors point out specific results about the “theoretical insights” claimed in line 59?
3. Could the authors abstract several unique insights on ”fine-grained” unlearning in the context of LLM?
4. Could the authors also discuss the computation-time cost of the existing framework compared with other methods?

**Ethical Concerns:**

["NO or VERY MINOR ethics concerns only"]

**Final Justification:**

Most of my concerns have been addressed, while I'm not fully convinced by the clarification about "theoretical insights" as there is no formal theoretical results, and most claims are supported with empirical evidence, while it doesn't affect my appreication on the work merits. In addition, it would also be better if the authors could discuss the practicality of the problem setting considered in this unlearning works and the real-world scenarios in the final version.

**Limitations:**

yes

**Quality:**

3

**Strengths And Weaknesses:**

Strengths:
1. This work consider the information-theoretic guidance for effectively measure the dependence on retain and forget data, and using SVD to identify principle directions in activation spaces are new (in the reviewer views) in unlearning.
2. The overall framework of FALCON is sounded and easy-to-understand, with intuitive illustrations.
3. The proposed method are generally demonstrated to be effective on various unlearning tasks and different LLMs.

Weaknesses:
Overall I think the current presentation is good for a new algorithmic proposal, while it could be better if the authors revise some part of clarification to make it clearer.
1. It is not very clear about the unlearning definition/target of the existing scenario, although the authors provide eq.1 as an general formulation which pursue two objectives like forgetting and retaining. The formal definition of unlearning in the context of LLM should be further elaborated, either on erasing specific knowledge within the LLM or just care about the specific output behavior.
2. It is a little bit hard to understand the fine-grained in the context of llm unlearning compared with the coarse-grained loss objective design in previous literature. To some extent, those mutual information-based reguarlization, SVD, and gradient conflicts analysis are also existing in previous literature of representation learning or exactly llm unlearning. It would be better if the authors highlight some new insights on the core intuition behind those technical combination.
3. Although the experimental results are promising on various unlearning tasks, could the authors also consider the efficiency of the existing methods compared with other baselines?
4. It is not easy to understand whether the proposed framework indeed approximate retrained model or realize the two pursuit in a special way. It could be better if the authors can further discuss this point.

---

> ### Author Rebuttal · Authors · 2025-07-29
>
> We appreciate the reviewer's feedback and support.
>
> For $\textbf{Question 1}$, we provide a detailed explanation below for your convenience:
>
> Coarse-grained unlearning approaches refer to existing methods that rely on simplistic loss combinations and empirical parameter selection strategies. These methods typically: (1) use grid search or empirical method to identify intervention parameters without principled guidance, (2) apply uniform modifications across broad parameter spaces without considering the specific knowledge distribution, and (3) employ random representation dispersion with uncontrolled gradient dynamics that struggle to precisely separate knowledge to be forgotten from knowledge to be retained. Such approaches often result in significant interference between forgetting and retention objectives, leading to degraded model utility or incomplete knowledge removal.
>
> In contrast, fine-grained unlearning as implemented in FALCON achieves more precise knowledge manipulation through: (1) Information-theoretic guidance that provides principled parameter selection by identifying layers with minimal knowledge entanglement using mutual information, (2) Targeted representation modification through our contrastive mechanism and Principal Offset Vectors that specifically steer activations away from unwanted knowledge directions identified via SVD, and (3) Regulated gradient dynamics via orthogonal projection that resolves conflicts between forgetting and retention objectives, minimizing damage to model utility. This fine-grained approach enables surgical knowledge removal with controlled interference, achieving superior balance between unlearning effectiveness and utility preservation compared to existing coarse-grained methods.
>
> We will provide a more detailed clarification and explanation of this distinction in the camera-ready version to ensure clarity for readers.
>
> For $\textbf{Question 2}:$
> Regarding the clarification on the specific theoretical insights claimed in line 59. Our approach provides the first principled, theoretically grounded method for parameter selection in LLM unlearning, moving beyond empirical grid search to a systematic framework based on knowledge entanglement quantification. The intuitive insights based on information-theoretic metrices are demonstrated through several key findings in our work:
>
> 1. Knowledge Distribution Patterns Across Model Architectures (Section 5.1.1, Figure 2): Our mutual information analysis reveals consistent patterns across different LLM architectures, showing that earlier layers exhibit lower MI values and more domain-specific representations, while deeper layers show higher entanglement. This provides intuitive understanding of how knowledge is distributed and organized within LLM architectures.
>
> 2. Gradient Conflict Correlation with Knowledge Entanglement (Figure 3): We demonstrate a strong relationship between mutual information and optimization conflicts - layers with low MI values exhibit significantly reduced gradient conflicts (cosine similarities near zero), while high-MI layers show pronounced, fluctuating conflicts. This validates MI as a reliable theoretical indicator for identifying optimal intervention parameters.
>
> These insights advance our intuitive understanding of knowledge representation and distribution in LLMs, providing a theoretical methodology for more principled unlearning approaches.
>
> For $\textbf{Question 3}:$
> We identify several critical insights that necessitate fine-grained approaches for LLM unlearning:
>
> 1. $\textbf{Knowledge Entanglement Complexity:}$ LLMs exhibit intricate knowledge entanglement patterns where forget and retain information are deeply intertwined across multiple layers and features. Unlike traditional machine learning models, LLMs store knowledge in distributed representations that cannot be effectively addressed through coarse-grained loss combinations or global parameter modifications.
>
> 2. $\textbf{Precision Requirements:}$ LLM unlearning demands precisely removing specific knowledge while preserving semantically related but distinct information. For example, forgetting a particular entity's biographical details while maintaining general knowledge about similar entities requires fine-grained manipulation of specific representational directions rather than broad parameter updates. However, the massive parameter space and complex transformer architectures of LLMs make naive approaches like full-parameter gradient ascent computationally prohibitive and prone to catastrophic forgetting. Fine-grained methods enable targeted interventions that maintain model utility while achieving effective knowledge removal.
>
> 3. $\textbf{Optimization Conflicts:}$ LLM unlearning inherently involves conflicting objectives between forgetting and retention that manifest at the gradient level. Fine-grained approaches like our gradient orthogonal projection can resolve these conflicts in specific parameter subspaces, while coarse-grained methods struggle with such optimization dynamics.
>
> These insights underscore why LLM unlearning requires sophisticated, fine-grained approaches that can navigate the complex knowledge structures and optimization landscapes inherent to large-scale language models.
>
> For $\textbf{Question 4}$:
> We have provided $\textbf{computational cost comparison between our method and other baselines in Appendix E.3 (line 821)}$, which demonstrates FALCON's competitive efficiency compared to other methods. Our results show that FALCON achieves 28.69 samples per second and completes 0.72 optimization steps per second, maintaining competitive performance with other unlearning methods while providing superior effectiveness. Additionally, to address your concern and validate the efficiency of our MI-guided approach, we have conducted specific analysis on the computational cost of MI-based parameter selection using the TOFU dataset. As shown in our MI cost analysis table, the parameter selection process scales reasonably with sample size, requiring only 67 seconds for 1\% of data up to 260 seconds for the full dataset. Importantly, this one-time parameter selection cost represents a small fraction of the total unlearning time, and the selected layers remain consistent across different sample sizes, demonstrating both the efficiency and stability of our MI-guided approach. Detailed discussion are shown at Review2, Question1 and new results will be integrated into camera ready version.
>
> MI Computational Cost Analysis (TOFU + Llama-3.2):
>
> Sample Size | Time(s)
> ------------|--------
> 10%         | 67
> 30%         | 79
> 50%         | 110
> 70%         | 165
> 100%        | 260

---

> > ### Comment · Reviewer_RCfm · 2025-08-05
> >
> > Thanks for the author's detailed response. Most of my concerns have been addressed, while I'm not fully convinced by the clarification about "theoretical insights" as there is no formal theoretical results, and most claims are supported with empirical evidence, while it doesn't affect my appreication on the work merits. In addition, it would also be better if the authors can discuss the practicality of the problem setting considered in this unlearning works and the real-world scenarios.

---

> ### Author Response · Authors · 2025-08-05
>
> We sincerely appreciate your timely feedback and constructive suggestions in improving our work:
>
> Regarding "theoretical insights" clarification:
> We acknowledge your point about the distinction between formal theoretical results and empirical insights. In the camera-ready version, we will revise the original statement “enabling principled parameter selection and providing **theoretical insights** into knowledge distribution across model architectures” to “providing **empirical insights** and principled parameter guidance for knowledge distribution across model architectures,” which more accurately reflects the nature of our contributions.
>
> Regarding the suggestion of discussing practicality of unlearning works in our manuscript:
> We will add the following discussion section to address the practical applications and real-world relevance of our work:
>
> The problem setting addressed by FALCON emerges from fundamental issues of current LLM deployment practices: the difficulty of selectively removing unwanted learned knowledge post-training to meet regulatory requirements such as GDPR's "right to be forgotten" [1]. Unlike traditional machine learning models where data can be simply excluded from future training cycles, LLMs encode knowledge in distributed representations across billions of parameters, making targeted removal extraordinarily challenging [2]. This creates a critical gap between learning capabilities and deployment needs for responsible AI systems. For instance, recent investigations revealed that major LLMs could generate detailed instructions for creating explosives and other dangerous materials when prompted appropriately, raising serious concerns about information safety [3]. When organizations deploy such models in real-world applications, previous solutions offer only binary choices: accept the safety risk or retrain the entire model at prohibitive cost. The development of LLM unlearning provides a third option, among which FALCON's fine-grained approach works by identifying which parameters encode the harmful knowledge through information-theoretic analysis, then using contrastive orthogonal unalignment to decouple that dangerous information while preserving the model's beneficial reasoning abilities. This precision becomes crucial when the same model must simultaneously forget specific harmful procedures while retaining general knowledge necessary for legitimate content, as emphasized by recent AI safety guidelines and industry best practices [2,4,5]. The practical significance extends beyond individual cases to systemic challenges: as LLMs become integral to enterprise workflows and public-facing applications, the ability to perform surgical knowledge modifications becomes essential infrastructure for responsible AI deployment, transforming unlearning from an academic curiosity into operational safeguards for maintaining both regulatory compliance and societal safety [6, 7].
>
> **Reference List:**
>
> [1] Ginart et al., "Making ai forget you: Data deletion in machine learning", Neurips 2019
>
> [2] Liu et al., "Rethinking machine unlearning for large language models", Nature Machine Intelligence 2025
>
> [3] Hendrycks et al., "Unsolved problems in ml safety".
>
> [4] Li et al., "The WMDP benchmark: Measuring and reducing malicious use with unlearning". ICML 2024
>
> [5] Martineau., "Why we’re teaching LLMs to forget things". IBM Research 2024
>
> [6] Dong et al., "Position: building guardrails for large language models requires systematic design", ICML 2024
>
> [7] Eldan et al., "Who's Harry Potter? Approximate Unlearning in LLMs", Microsoft Research 2023
>
> We hope these revisions adequately address your concerns and enhance the clarity and impact of our work. If you have any additional questions or suggestions for further improvement, we would be delighted to discuss them.

---

### Official Review · Reviewer_aqr3 · 2025-07-02

**Clarity:** 2
**Significance:** 2
**Originality:** 2
**Rating:** 4
**Confidence:** 4

**Summary:**

Existing training-time unlearning methods rely on coarse-grained loss combinations. This work proposes a representation-guided framework for targeted knowledge removal. It utilizes information-theoretic guidance to identify layers with minimal entanglement and employs contrastive mechanisms to enhance the separation of representations.

**Questions:**

1. **Robustness to Knowledge Recovery:** The reason why the proposed method is robust against knowledge recovery techniques needs more discussion. It would be helpful to include a more detailed and intuitive explanation or theoretical analysis. Alternatively, comparing with other strong unlearning baselines under these attack methods could better demonstrate the effectiveness of the proposed defense.
2. **Evaluation Metrics in Table 2:** In Table 2, FALCON seems to achieve the second-lowest forget metric score. Have you defined a new criterion for determining which method performs best?
3. The first issues for current research is that “existing approaches typically rely on empirical methods like grid search to identify intervention parameters”, could you give several related work?
4. Minor part: In Table 3, for forget01, NPO seems to achieve the best MU.

**Ethical Concerns:**

["NO or VERY MINOR ethics concerns only"]

**Final Justification:**

The concerns have been addressed. I would like give 4 to this paper.

**Limitations:**

yes

**Quality:**

2

**Strengths And Weaknesses:**

# Strengths

- Introduces a new fine-grained framework for LLM unlearning.
- The use of mutual information-based guidance and gradient orthogonal projection enhances the interpretability and robustness of the unlearning process.
- Provides extensive experiments on three commonly used benchmarks and multiple LLMs.

# Weaknesses

- **Validation of Layer Selection Metric on TOFU:** To validate the metric used for identifying layers with minimal entanglement, visualizations and analysis of gradient conflicts on the TOFU dataset should be provided. Since the forget and retain sets in this dataset are relatively similar, demonstrating that mutual information can disentangle knowledge related to similar QA pairs would strengthen the argument.
- **Baselines:** For the TOFU dataset, since retained data is used for FALCON, the retained version of DPO should be included as a baseline, not just IdkDPO. Additionally, it is unclear whether the version of NPO used is the original or one of the extended variants (NPO-RT or NPO-KL).
- **Limited Experiments:**
    - The MUSE benchmark appears to only report results on the “book” subset. Results on the “news” subset should also be included.
    - The TOFU dataset experiments are limited to only one model (LLaMA-3.2-1B-Instruct), whereas the original paper provides results for finetuned and retained models based on LLaMA-2-7B. The same problem for WMDP. These limitations make it unclear whether the proposed method can generalize to other model architectures or larger models.

---

> ### Author Rebuttal · Authors · 2025-07-29
>
> Thanks for your comments and we fully understand your concerns. To provide clear responses to your feedback, we use W to denote weakness and Q to denote question to address each concern below. $\textbf{The reference list please follow Reviewer 1 due to text limit.}$
>
> $\textbf{W1: }$
>
> To mitigate your concern, we are pleased to provide additional experimental analysis on TOFU benchmark [9] to demonstrate the efficiency and practicality of our MI-guided method. We evaluated the computational overhead of MI estimation across different sample sizes while maintaining PCA dimensionality reduction at 95% variance retention to ensure computational feasibility. The results below show computational efficiency across varying sample proportions ranging from 10% to the full dataset and exhibit consistent results.
>
> MI-guided Method Cost Analysis (TOFU + Llama-3.2):
>
> Sample Size | Time(s) | Optimal Parameter Layer
> ------------|---------|------------------------
> 10%         | 67      | 3
> 30%         | 79      | 3
> 50%         | 110     | 3
> 70%         | 165     | 3
> 100%        | 260     | 3
>
> Furthermore, we provide the normalized MI results on the full sample size below (lower is better) to give an intuitive understanding of our method. Due to rebuttal requirements that prevent figure inclusion, we present these results in tabular form here and will provide visualizations in the camera-ready version.
>
> Layer         | 0    | 1    | 2    | 3    | 4    | 5    | 6    | 7    | 8    | 9    | 10   | 11   | 12   | 13   | 14   | 15
> --------------|------|------|------|------|------|------|------|------|------|------|------|------|------|------|------|------
> Normalized MI | 0.78 | 0.43 | 0.19 | 0.00 | 0.18 | 0.07 | 0.05 | 0.14 | 0.23 | 0.41 | 0.51 | 0.65 | 0.65 | 0.68 | 0.80 | 1.00
>
> The aforementioned results demonstrate: (1) MI computation scales reasonably with sample size, maintaining manageable computational overhead; (2) layer selection remains consistent across different sample sizes, indicating stability of the MI-guided approach; (3) the method successfully identifies optimal intervention layers, with Layer 3 indeed being the parameter selection in accordance with our best performance in our experimental results on TOFU, even forget and retain sets have high similarity.
>
> These results confirm that MI-guided parameter selection is both computationally feasible and scalable, providing a principled foundation for efficient unlearning. Combined with our original computational efficiency analysis in Appendix E.3, both the MI-guided parameter selection and our method's inherent efficiency demonstrate the scalability of our approach.
>
> $\textbf{W2:}$
>
> Regarding baselines, our experimental setup strictly follows two widely community-recognized frameworks: the TOFU benchmark [9] and the OpenUnlearning framework [5].
>
> DPO Clarification: Our implementation includes IdkDPO (often abbreviated as DPO in some papers), which utilizes standard DPO with specialized preference pair construction for unlearning context. IdkDPO sets preferred responses to "I don't know" for forget queries while treating original answers as rejected responses. This expression was initially introduced in TOFU paper and adopted as a core baseline in OpenUnlearning.
>
> NPO Clarification: Our experiments utilized the NPO implementation from the OpenUnlearning framework as specified by Zhang et al. [7], which employs the negative feedback term for forget data combined with a standard retain loss term.
>
> Both frameworks represent community consensus on standardized evaluation protocols. TOFU established IdkDPO as a basic baseline for entity unlearning, while OpenUnlearning provides unified implementations of core methods. Our adherence ensures reproducibility and fair comparison with established benchmarks in the literature.
>
> $\textbf{W3:}$
>
> To clarify, the results presented in our paper is MUSE News. However, for results across different MUSE subsets, we are happy to provide additional results on the MUSE Books below to address this concern and it will be integrated into the final version for a comprehensive evaluation.
>
> Method     |F_knowmem|F_verbmem|R_knowmem
> -----------|--------------|--------------|---------------
> Finetuned  |0.47          |1.0           |0.69
> Retain     |0.3           |0.14          |0.69
> GradAscent |0             |0             |0
> GradDiff   |0.18          |0.16          |0.3
> NPO        |0.32          |0.84          |0.55
> SimNPO     |0.32          |0.84          |0.54
> RMU        |0.29          |0.79          |0.48
> FALCON     |0.06          |0.05          |0.68
>
> For model selection on TOFU, we follow established unified benchmark OpenUnlearning while demonstrating generalization across different architectures. For MUSE we still used LLaMA-2-7B series following standard configurations, while for TOFU we deliberately employed LLaMA-3.2 series as provided by the OpenUnlearning to demonstrate the reviewer's concerned generalizability across diverse model architectures from established to latest versions. For WMDP [1], we conducted comprehensive experiments on Zephyr-7B, the primary model used in the WMDP paper, where we reproduced all baselines for fair comparison. While hardware limitations prevent us from testing on the larger models shown in their supplementary experiments, we selected same architectures including Yi and Mistral series used in WMDP papers to demonstrate our method's generalizability across different model families. This cross-architectural validation validates FALCON's broad applicability, and it is worth noting that most established unlearning works [2, 6] are also primarily evaluated on models of similar scale (7B and below), reflecting the community's standard practice for unlearning evaluation.
>
> $\textbf{Q1:}$
>
> FALCON's robustness against knowledge recovery stems from creating fine-grained representational changes rather than random directions. Our contrastive mechanism with Principal Offset Vectors systematically steers activations away from the dominant subspaces associated with unwanted knowledge, while orthogonal gradient projection ensures updates move away from retention-sensitive directions. This approach intuitively creates targeted geometric separation in the model's representational space, making forgotten knowledge structurally inaccessible rather than relying on random perturbations. Our comparative evaluation against Enhanced GCG attacks demonstrates that while existing baselines like RMU [1] show vulnerability to increasing attack iterations, FALCON maintains consistent resistance across attack intensities [10]. Furthermore, our logit lens analysis in Section 5.3 also empirically reveals that unlearned knowledge remains inaccessible across different architectural components, validating that our method achieves persistent internal representational changes. Moreover, we  provide additional intuitive comparative results with other strong baselines below to address your concerns. Under identical jailbreaking attacks, RMU's knowledge can be substantially recovered while FALCON maintains resistance.
>
> Method        |Original|Unlearned|Recovery
> --------------|--------|---------|---------------
> RMU (Bio)     |65.4    |50.8     |58.5 ± 3.2
> FALCON (Bio)  |65.4    |27.7     |28.1 ± 0.5
> RMU (Cyber)   |42.6    |33.5     |41.8 ± 2.1
> FALCON (Cyber)|42.6    |25.3     |25.5 ± 0.8
>
> $\textbf{Q2 and 4:}$
>
> We understand your confusion regarding evaluation criteria and would like to clarify the unlearning evaluation. Unlearning evaluation requires holistic assessment $\textbf{beyond forget metrics alone, as effective unlearning must balance knowledge removal with utility preservation.}$ While FALCON achieves very low forget scores (0.02 and 0.03), our key strength lies in superior retain performance (0.54). This balance is crucial because methods like GradAscent achieve perfect forget scores (0.00) but completely fail to preserve useful knowledge (retain score: 0.00), rendering them impractical for deployment. Comprehensive metric explanations are in Appendix D.3.
>
> Regarding NPO's slightly higher Model Utility (0.56 vs 0.55) in Forget01, FALCON demonstrates superior Forget Quality (0.99 vs 0.92), indicating more effective knowledge removal. As forget dataset size increases to Forget05 and Forget10, FALCON consistently outperforms baselines with optimal balance, achieving the highest MU (0.60) in Forget10 while maintaining effective forgetting. These results demonstrate FALCON's scalability across varying dataset complexities and architectures. We will update this clarification in the camera-ready version.
>
> $\textbf{Q3:}$
>
> We will include the citations and discussion in the camera-ready version based on your valuable suggestions. Early methods like GradAscent [11] typically perform full-parameter optimization, which can be computationally expensive and lead to catastrophic forgetting. Selective parameter modification approaches, including RMU [1] and task vector methods [8], have demonstrated that targeted parameter updates can effectively address the limitations of full-parameter optimization and have contributed this insight to subsequent unlearning methods. However, these methods still rely on empirical techniques such as extensive grid search across model layers to obtain their observations regarding optimal parameter selection [1]. Despite demonstrating the effectiveness of selective parameter modification, the fundamental limitation remains that these methods lack principled theoretical guidance for parameter selection, as model architectures, knowledge entanglement patterns, and task-specific conditions vary significantly across different scenarios. This highlights the necessity for our MI-guided approach that provides interpretable, theory-driven parameter identification.

---

> > ### Comment · Reviewer_aqr3 · 2025-08-05
> >
> > Sorry for the late reply. I have a few additional questions and clarifications.
> >
> > For Weak 2, access to retain data plays an important role [1], particularly for the TOFU datasets. Since FALCON utilizes both retain and forget data, adding experiments for the *retained* version of DPO would be a fair comparison in my view, as done in [2]. Similarly, if FALCON uses only IDK data (no retain), then comparing with IDK-DPO would be a fairer baseline.
> >
> > Could the authors also provide a more detailed discussion of the MUSE benchmark? I did not see an explicit explanation in Section 5.2.1 of why FALCON outperforms the baseline. It would be interesting to understand how FALCON can effectively retain relevant content with high KnowMem scores while still achieving nearly complete forgetting on the KnowMem-forget set.
> >
> > In addition, what is the unlearning setting used for the MUSE benchmark? The reported results for NPO_GDR (the provided table) show a large gap compared to the numbers in the original paper [3].
> >
> > Regarding Weak 3.2, on the TOFU dataset, the paper only reports results for the LLaMA series, whereas the original TOFU paper also includes results for the Phi model. Since FALCON relies on manipulating representations, it is important to assess its effectiveness across different model architectures, particularly on datasets where the retain and forget data share similar patterns (such as TOFU), as demonstrated in [4], in my view.
> >
> > Thanks for the author’s response. Please see these questions.
> >
> > *[1] LLM Unlearning via Loss Adjustment with Only Forget Data*
> >
> > *[2] Reversing the Forget–Retain Objectives: An Efficient LLM Unlearning Framework from Logit Difference*
> >
> > *[3] MUSE:MachineUnlearningSix-Way Evaluation for Language Models*
> >
> > *[4] LUNAR:LLMUnlearning via Neural Activation Redirection*

---

> > > ### Author Response · Authors · 2025-08-07
> > >
> > > We deeply appreciate your efforts in continued engagement with our work. We will use **Q** to index your questions for your convenience.
> > >
> > > **Q1**
> > >
> > > We appreciate the opportunity to clarify our experimental setup. Upon re-examining the OpenUnlearning's DPO implementation, we can confirm that the DPO utilizes both forget and retain data, identical to our method, ensuring fair comparison. Specifically, The implementation applies preference optimization on forget data through dpo_loss while simultaneously employing retain_loss to maintain retain data performance, with the final objective being loss = γ × dpo_loss + α × retain_loss. This confirms that both FALCON and the DPO baseline operate under identical data conditions.
> > >
> > > **Q2**
> > >
> > > FALCON's ability to achieve high r-KnowMem scores while achieving nearly complete knowledge removal on MUSE stems from its fine-grained knowledge disentanglement capability. For instance, in the MUSE News task, the challenge lies in removing specific copyrighted articles while preserving general journalistic knowledge. FALCON addresses this through MI-guided parameter selection, which identifies parameters where representations of specific news content and general journalistic knowledge are least entangled, enabling minimal parameter modifications compared to baseline methods that apply global updates across all layers. The Principal Offset Vectors then perform targeted representational unalignment by using SVD to identify dominant principal components capturing memorized article-specific patterns, while orthogonal gradient projection resolves optimization conflicts to prioritize preservation of general knowledge while steering forget-related activations away from principal subspaces containing copyrighted details. This selective parameter intervention combined with conflict-resolved gradient updates minimizes disruption to broader capabilities, maintaining general performance while achieving fine-grained knowledge separation that other coarse-grained methods cannot accomplish.
> > >
> > > **Q3**
> > >
> > > Regarding the NPO result, our experimental setup strictly follows the OpenUnlearning's standardized implementation. As discussed in MUSE repository issue #2, there are recognized implementation differences between the MUSE paper and the original NPO paper's implementation, such as discrepancies in loss function construction and logits computation. Multiple developers confirmed that the original MUSE code contains inconsistencies may lead to unstable training behavior with excessive forgetting and uncontrolled preference optimization. The OpenUnlearning developers addressed these inconsistencies by adopting a corrected version aligning with the original NPO paper's theoretical foundation. We strictly adhered to the OpenUnlearning's standardized settings for fair comparison, including learning rate (1e-5), optimizer (paged_adamw_32bit), DeepSpeed ZeRO Stage 3 distributed training on 3×A100 GPUs, and unified data preprocessing/evaluation pipeline.
> > >
> > > **Q4**
> > >
> > > We fully understand your suggestion regarding cross-architecture generalizability. We conducted supplementary experiments on Phi-1.5 using TOFU to address this concern.
> > >
> > > When forgetting 5% of the data, FALCON achieves superior performance by maintaining high model utility (0.39) while achieving effective forgetting (0.88), outperforming other methods that either sacrifice utility or fail to achieve adequate forgetting.
> > >
> > > | Method     | FQ      | MU   |
> > > |------------|---------|------|
> > > | Original   | 0.0     | 0.4  |
> > > | GradAscent | 2.56e-7 | 0.22 |
> > > | GradDiff   | 0.04    | 0.32 |
> > > | DPO        | 0.32    | 0.10 |
> > > | NPO        | 0.01    | 0.36 |
> > > | RMU        | 0.25 | 0.26 |
> > > | FALCON     | 0.88    | 0.39 |
> > >
> > > When forgetting 10% of the data, FALCON shows high consistency, maintaining both effective forgetting (0.90) and model utility (0.40), while baseline methods show severe utility degradation or complete failure.
> > >
> > > | Method     | FQ       | MU   |
> > > |------------|----------|------|
> > > | Original   | 0.0      | 0.4  |
> > > | GradAscent | 2.56e-20 | 0.0  |
> > > | GradDiff   | 2.73e-12 | 0.26 |
> > > | DPO        | 6.92e-5  | 0.07 |
> > > | NPO        | 1.49e-16 | 0.31 |
> > > | RMU        | 5.81e-14 | 0.29 |
> > > | FALCON     | 0.90     | 0.40 |
> > >
> > > Furthermore, we place great emphasis on generalizability in our original manuscript. Our comprehensive experimental design demonstrates FALCON's effectiveness and practicality across the LLM unlearning domain, spanning multiple tasks including harmful knowledge removal (WMDP), entity unlearning (TOFU), and copyrighted content unlearning (MUSE), as well as diverse model series including Zephyr, Mistral, LLaMA-2 and LLaMA-3. This extensive cross-task and cross-architecture validation confirms our method's generalizability and we hope this evidence could satisfy your expectations.
> > >
> > > Following your valuable suggestions, we will update the discussed content in the camera-ready version, including the experiments, discussions, and citations you have suggested.

---

> > > > ### Comment · Reviewer_aqr3 · 2025-08-07
> > > >
> > > > Thanks for your detailed discussion and the experiments. These explanations are helpful. My concerns have been addressed. So I am raising my score to 4. Thanks so much for your patient and  detailed response. Good luck.

---

> > > > > ### Author Response · Authors · 2025-08-07
> > > > >
> > > > > Thank you for your valuable comments and recognition of our work. We greatly appreciate your patience and detailed feedback throughout the rebuttal process.

---

### Official Review · Reviewer_GXzf · 2025-07-02

**Clarity:** 2
**Significance:** 3
**Originality:** 3
**Rating:** 4
**Confidence:** 3

**Summary:**

The paper introduces FALCON (Fine-grained Activation Manipulation by Contrastive Orthogonal Unalignment), a novel framework for selective knowledge unlearning in large language models (LLMs). FALCON addresses key challenges in machine unlearning, such as precise knowledge separation, preservation of model utility, and resistance to knowledge recovery attempts.

**Questions:**

1. Please refer to the Weakness section.
2. What are the advantages and disadvantages of using task vectors [1] or negative preference optimization [2] to solve toxicity/unlearning problems compared to the methods mentioned in the paper.

[1] Editing Models with Task Arithmetic. ICLR 2023.

[2] Simplicity Prevails: Rethinking Negative Preference Optimization for LLM Unlearning. Arxiv 2024.

**Ethical Concerns:**

["NO or VERY MINOR ethics concerns only"]

**Limitations:**

yes

**Quality:**

3

**Strengths And Weaknesses:**

Strengths
1. The paper presents a well-designed framework (FALCON) with a clear theoretical foundation (mutual information, contrastive learning, gradient projection). The use of SVD for principal directions and gradient orthogonalization is technically sound.
2. FALCON could be used in cross-domain generalizability assessment beyond harmful content, which is valuable.

Weaknesses
1. Experiments are conducted on relatively small LLMs (≤7B parameters). The scalability to larger models remains unverified.
2. The granularity of parameter selection is not very clear. The gradient should be calculated on each parameter matrix or neuron, but it seems that parameter selection is expressed as the granularity of layers.
3. The use of two basic models and baseline systems in Section 5.1 and 5.2 is confusing, and there should at least be some common baselines for comparison.

---

> ### Author Rebuttal · Authors · 2025-07-29
>
> We appreciate the reviewer's feedback and happy to address each concern one by one below.
>
> $\textbf{For Weakness 1 and 3:}$
>
> Thanks for pointing out your concern about experiments on models and baseline methods. Our experimental design is carefully aligned with established practices in the machine unlearning literature and represents current state-of-the-art evaluation standards.
>
> The model selection strictly follows best practices established by leading works in this field. For example, RMU [1], the current state-of-the-art representation-guided adaptation method, conducted their core experiments on Zephyr-7B. We not only replicated their experimental setup but comprehensively evaluated all baseline methods they employed, including LLMU [2], SCRUB [3], and SSD [4], ensuring fair and meaningful comparisons across WMDP benchmark with RMU.  In WMDP benchmark experiments, we employed both Yi-series and Mistral-series models (Appendix E.1) to demonstrate our method's generalization for multiple-architecture evaluation, which are also the primary architectures evaluated in the RMU and WMDP paper.
>
> For entity unlearning and copyrighted content unlearning tasks, we adopted the latest OpenUnlearning framework [5], which represents the community's recognition on unified evaluation standards. It is also worth noting that most existing unlearning methods in the literature, such as LLMU [2] and SimNPO [6], are also evaluated on less than 7B-scale models due to the hardware and resource constraints of LLM unlearning. The models we employed (e.g., LLaMA-3.2, LLaMA-2-7B) are the standard evaluation models recommended by this OpenUnlearning framework and community, ensuring reproducibility and generalization of our method. We also provided extensive evaluations with widely recognized baselines like NPO [7], covering entity unlearning and copyrighted content unlearning tasks comprehensively. FALCON's core innovations are theoretically model-agnostic. Our approach targets the intrinsic representational structures of LLMs rather than specific architectural details, providing a foundation for scaling to larger models. Given hardware limitations, maintaining experimental consistency with SOTA methods and utilizing community-recognized unified evaluation frameworks provides sufficient evidence for FALCON's efficacy.
>
> $\textbf{For Weakness 2:}$
>
> We clarify that our parameter selection is guided by mutual information rather than gradients. We compute $I(F_l; R_l)$ where $F_l$ and $R_l$ represent the activations (intermediate hidden representations) at layer $l$ for forget and retain datasets, respectively. Since activations are naturally layer-level outputs in transformer architectures, layer-level granularity for MI-based selection is theoretically appropriate. The gradient conflicts shown in our experiments serve as intuitive empirical evidence that layers with lower MI indeed facilitate better knowledge disentanglement. When MI is minimal between forget and retain representations, the gradient conflicts between forgetting and retention objectives are also reduced, enabling more effective decoupling of knowledge to be removed versus preserved. Subsequently, we perform parameter updates within the selected optimal layer l indeed targeting specific parameter matrices, normally focusing on MLP matrices like other leading works [1]. This choice is motivated by the fact that MLP matrices, particularly the output projection weights, control the final knowledge output of each layer and directly influence information propagation to subsequent layers, while being relatively safe to modify without overly disrupting the model's fundamental representational capabilities.
>
> $\textbf{For Question 2:}$
>
> Task vectors [8] offer computational efficiency through arithmetic operations on model weights, enabling lightweight unlearning without requiring extensive retraining. Their modular approach allows for easy composition and reversal of edits. However, task vectors suffer from significant limitations: they oversimplify knowledge structures through linear assumptions that fail to capture complex knowledge entanglement in LLMs, lack fine-grained control over specific knowledge removal, and may inadequately address deeply embedded representations that require non-linear transformations.
>
> Negative Preference Optimization [7] reformulates unlearning as preference learning, leveraging established preference optimization techniques. NPO can handle diverse unlearning scenarios through flexible preference specification. Nevertheless, NPO is prone to catastrophic collapse when forget and retain distributions exhibit high similarity, lacks fine-grained knowledge localization control, and may struggle with precise separation of entangled knowledge representations.
>
> To better mitigate your concern between our method and existing mainstream baselines, we also provide a more comprehensive discussion that will be integrated into the Related Work section and Appendix in the camera-ready version to address your concerns as follows:
>
> Existing LLM unlearning approaches operate through distinct methodological paradigms with inherent trade-offs. Methods like Task Vectors [8] use arithmetic operations on model weights for computational efficiency but oversimplify knowledge structure through linear assumptions that fail to capture complex knowledge entanglement. Gradient-based approaches including LLMU [2] and GradAscent [11] apply gradient ascent to forgotten knowledge but frequently induce catastrophic forgetting and optimization instability. Preference optimization methods such as NPO and its variants reformulate unlearning as preference learning, yet are prone to catastrophic collapse when distributions exhibit high similarity and lack fine-grained knowledge localization control [6,7]. Representation manipulation methods like RMU [1] modify intermediate activations using MSE loss and random unit vectors for steering, offering more targeted intervention than gradient methods but relying on empirical layer selection and lacking targeted separation mechanisms.
>
> Similar to RMU, FALCON belongs to the representation-based paradigm, optimizing only a subset of intermediate layers and parameters for effective LLM unlearning. However, FALCON advances through three key innovations: information-theoretic parameter guidance using mutual information to identify optimal intervention layers, targeted representation disentanglement via contrastive learning with Principal Offset Vectors for precise knowledge separation, and gradient orthogonal projection for conflict resolution between objectives. Compared to RMU's MSE-based random steering with heuristic selection, our contrastive mechanism provides more principled and targeted representation manipulation, achieving superior unlearning effectiveness while preserving model utility.
>
> $\textbf{Reference List for Reviewer 1 and Reviewer 2:}$
>
> [1] Li et al., "The WMDP benchmark: Measuring and reducing malicious use
> with unlearning". ICML 2024
>
> [2] Yao et al., "Large Language Model Unlearning". Neurips 2024
>
> [3] Kurmanji et al., "Towards unbounded machine unlearning". Neurips 2024
>
> [4] Foster et al., "Fast machine unlearning without retraining through selective synaptic dampening", AAAI 2024
>
> [5] Dorna et al., "OpenUnlearning: Accelerating LLM Unlearning via Unified Benchmarking of Methods and Metrics".
>
> [6] Fan et al., "Simplicity Prevails: Rethinking Negative Preference Optimization for LLM Unlearning", Neurips SafeGenAi 2024
>
> [7] Zhang et al., "Negative Preference Optimization: From Catastrophic Collapse to Effective Unlearning", COLM 2024
>
> [8] llharco et al., "Editing Models with Task Arithmetic". ICLR 2023.
>
> [9] Maini et al., "TOFU: A task of fictitious unlearning for LLMs". COLM 2024
>
> [10] Łucki et al., "An Adversarial Perspective on Machine Unlearning for AI Safety". TMLR 2024.
>
> [11] Jang et al., "Knowledge Unlearning for Mitigating Privacy Risks in Language Models". ACL 2023.

---

### Decision · Program_Chairs · 2025-09-17

**Decision:**

Accept (poster)

**Comment:**

This paper focuses on LLM unlearning through activation manipulation. It aims to improve existing unlearning methods by addressing the limitations of relying on coarse-grained loss combinations, and instead proposes a representation-guided framework for targeted knowledge removal. The approach leverages information-theoretic guidance to identify layers with minimal entanglement and employs contrastive mechanisms to enhance representation separation.

All five reviewers gave this paper positive scores, highlighting its effective framework with solid theoretical foundation and extensive experiments. One main initial concern raised by several reviewers was whether the baselines were set up correctly and whether the selected baselines were sufficient. During the discussion stage, most of these concerns from the reviewers were addressed.